# Single gene analysis in yeast suggests nonequilibrium regulatory dynamics for transcription

Robert Shelansky[1], Sara Abrahamsson[2], Christopher R. Brown[1,3], Michael Doody[1], Tineke L. Lenstra [4], Daniel R. Larson [5] & Hinrich Boeger [1]✉

Fluctuations in the initiation rate of transcription, the first step in gene expression, ensue from the stochastic behavior of the molecular process that controls transcription. In steady state, the regulatory process is often assumed to operate reversibly, i.e., in equilibrium. However, reversibility imposes fundamental limits to information processing. For instance, the assumption of equilibrium is difficult to square with the precision with which the regulatory process executes its task in eukaryotes. Here we provide evidence − from microscopic analyses of the transcription dynamics at a single gene copy of yeast − that the regulatory process for transcription is cyclic and irreversible (out of equilibrium). The necessary coupling to reservoirs of free energy occurs via sequence-specific transcriptional activators and the recruitment, in part, of ATP-dependent chromatin remodelers. Our findings may help explain how eukaryotic cells reconcile the dual but opposing requirements for fast regulatory kinetics and high regulatory specificity.

Eukaryotic transcriptional activators promote the transcription of specific genes by virtue of their ability to bind gene-specific DNA sequences (enhancers) and recruit a host of proteins that either pertain to the transcription machinery or facilitate its assembly at the transcription start site, including ATP-dependent chromatin remodelers[1,2]. It is generally assumed that the transcriptional output of a gene is determined by the equilibrium occupancy of its enhancers by activators[3–5].

Activators find their target sequences by trial and error and thus bind DNA also non-specifically. For eukaryotic activators, the energetic differences between specific and non-specific sequence binding are surprisingly small[6–8], small enough to be easily erased by the abundance of non-specific activators. How the eukaryotic cell solves this specificity problem is unknown.

Specific binding events differ from non-specific ones by a statistically longer residence time of the activator on the DNA[8]. A potential solution to the specificity problem, therefore, is to respond to activator residence time rather than occupancy, a feat that might be accomplished by kinetically proofreading activator identity[9,10]. The kinetic proofreading mechanism increases specificity by exploiting the kinetic difference between specific and non-specific DNA binding twice, before and after activator-dependent removal of a transcriptional repressor which delays advance to the transcriptionally active state[10]. Repressor removal requires energy, which is provided not by the surrounding heat bath but by an external work reservoir, avoiding the entropic cost of converting heat into work[10–12]. The expended free energy is eventually dissipated into the heat bath as the gene returns to its thermodynamically preferred repressed state, closing an irreversible reaction cycle where all microscopic paths in the same direction of rotation, say clockwise, are more probable than their time-reversed, anticlockwise counterparts (*cf.* Figure 1a). The gene, thus, is driven in and out of states that support transcription. As a consequence, anticlockwise paths, *i.e.*, paths that bypass the proofreading delay step, are discouraged.

[1]Department of Molecular Cell and Developmental Biology, University of California, Santa Cruz, CA, USA. [2]Department of Electrical and Computer Engineering, University of California, Santa Cruz, CA, USA. [3]Korro Bio, Cambridge, MA, USA. [4]Division of Gene Regulation, The Netherlands Cancer Institute, Oncode Institute, Amsterdam, The Netherlands. [5]Laboratory of Receptor Biology and Gene Expression, National Cancer Institute, National Institutes of Health, Bethesda, Maryland, USA. ✉e-mail: hboeger@ucsc.edu

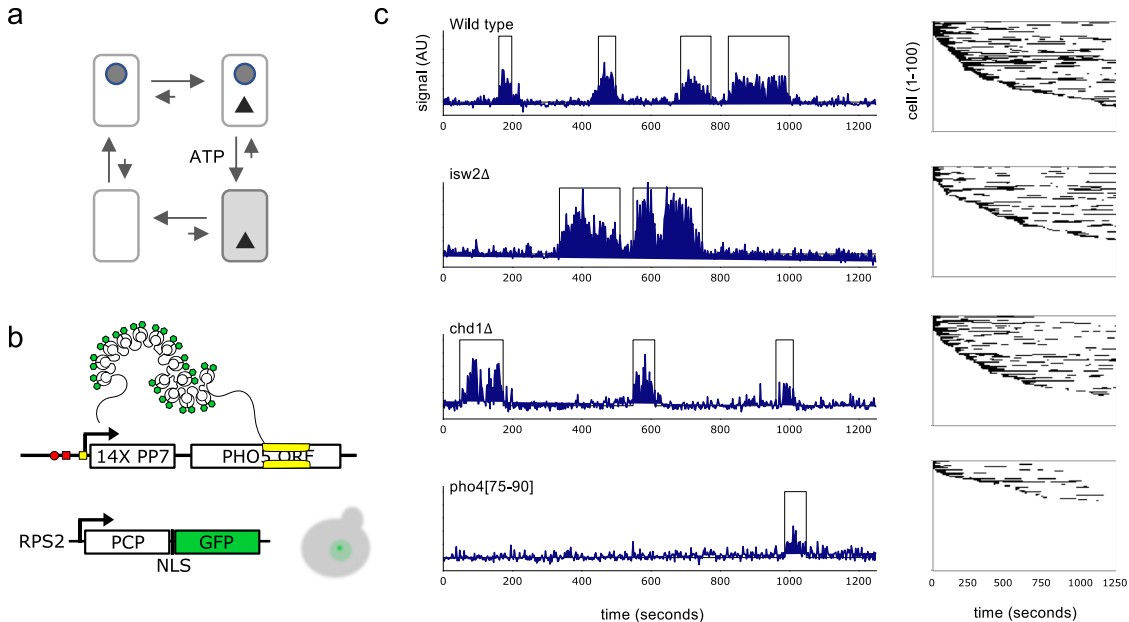

**Fig. 1 | In steady state, periods of transcriptional activity (ON) are interrupted by periods of inactivity (OFF). a** Graph for the kinetic proofreading of activator identity[10]. The promoter is depicted as a box, the bound transcriptional activator by a black triangle, a repressor (*e.g.*, nucleosome) whose removal (delay step) is required for transcription by a gray dot. Arrows (directed edges) indicate allowed transitions between microstates. Different arrow lengths indicate that clockwise transitions are statistically preferred over counterclockwise transitions due to the transfer of free energy to the system in the transition from repressed to derepressed promoter in the presence (but not absence) of the activator and its dissipation upon return to the repressed state. **b** Scheme of the modified *PHO5* gene (top) and chromophore gene (bottom). *PHO5* contained a tetradecamer of the

binding sequence for the PP7 coat protein (PCP). PCP was expressed as a fusion with a green fluorescent protein (GFP) with nuclear localization signal (NLS) under the control of the *RPS2* promoter (bottom). The cartoon of budding cells (in gray) shows that nuclear GFP is concentrated at the site of *PHO5* transcription. **c** Left panel column: Examples of single-cell fluorescence time series for wild type, *isw2Δ*, *chd1Δ*, and *pho4Δ[75–90]*. Right panel column: Each box contains 100 binary time series obtained from change point detection analysis of fluorescent time series and ordered in the sequence of increasing time between the beginning of the observational time window and the onset of the first ON period. Horizontal black lines indicate the length of ON periods, white lines the length of OFF periods, which often spanned the entire width of the observational time window.

The thermodynamics of activator-DNA binding imposes an upper bound to the specificity of transcriptional regulation. In equilibrium, the regulatory process approaches this bound as the activator concentration tends to zero; the time between transcription events, then, tends to infinity; the regulatory process becomes slow and increasingly erratic[10]. Away from equilibrium, kinetic proofreading permits regulatory specificities close to the upper bound of specificity, including values above that bound, at activator concentrations that afford fast regulatory kinetics and low transcription noise[10].

The nucleosome appears to be ideally suited to assume the task of repressor in the kinetic proofreading scheme (nucleosomal proofreading hypothesis)[10]. The nucleosome is a ubiquitous repressor of nuclear gene transcription[13]. Promoter nucleosomes evidently partake in the random dynamics of the regulatory mechanism for transcription, for genes assume alternative promoter nucleosome configurations in activating conditions, including the nucleosome-free and fully nucleosomal promoter[14]. Gene regulation entails the unwinding of promoter nucleosomes[15]. Nucleosome unwinding, for most DNA sequences, requires a significant amount of work, $\Delta G^{o'} \approx 44\,k_B T$ per nucleosome on the assumption that nucleosomes were unwound reversibly[16]. Thermodynamics, therefore, favors nucleosome formation over a wide range of histone concentrations. In this event, nucleosome unwinding elevates the system's free energy, while rewinding dissipates this energy into the surrounding heat bath[10,11]. There is little doubt that nucleosomes are unwound by ATP-dependent chromatin remodelers that are recruited to promoters by activators[17–20].

It is unknown, however, whether the nucleosome dynamics of transcriptionally induced promoters are driven away from equilibrium

in this or any other way. Both the wrapping and unwrapping of nucleosomes are catalyzed by ATP-dependent chromatin remodelers[21]. ATP may be expended to overcome high energy barriers and thus accelerate the approach to equilibrium rather than maintain a non-equilibrium steady-state[3].

Most data appear to conform to the expectations of a regulatory process in equilibrium[3]. It is possible, however, that this conformity is the result of inevitable, and often also deliberate, coarse-graining. Most microstates of a system remain indistinguishable to the observer who therefore lumps them together into observable mesostates. Observational coarse-graining may easily generate a false impression of reversibility[22]. For instance, an irreversible (nonequilibrium) steady-state process requires that its reaction network (graph) is cyclic; when coarse-graining reduces the cyclic to an acyclic graph, the steady-state process becomes reversible[11,23]. (Thus, by virtue of its non-cyclic transition graph, the standard two-state random-telegraph model of gene regulation[24] implies equilibrium dynamics.) Furthermore, predicted differences between reversible and irreversible processes are often subtle or otherwise difficult to measure[5].

Here we provide evidence that the regulatory process of *PHO5* transcription in yeast, a classic paradigm for the analysis of transcriptional regulation by chromatin structure[25], breaches a barrier imposed by the detailed balance conditions for equilibrium.

## Results
### The biological model
Transcriptional activation of *PHO5* requires binding of the basic helix-loop-helix factor Pho4 to its enhancer UASp1[26]. Pho4 binding increases the probability of promoter states with fewer nucleosomes[14],

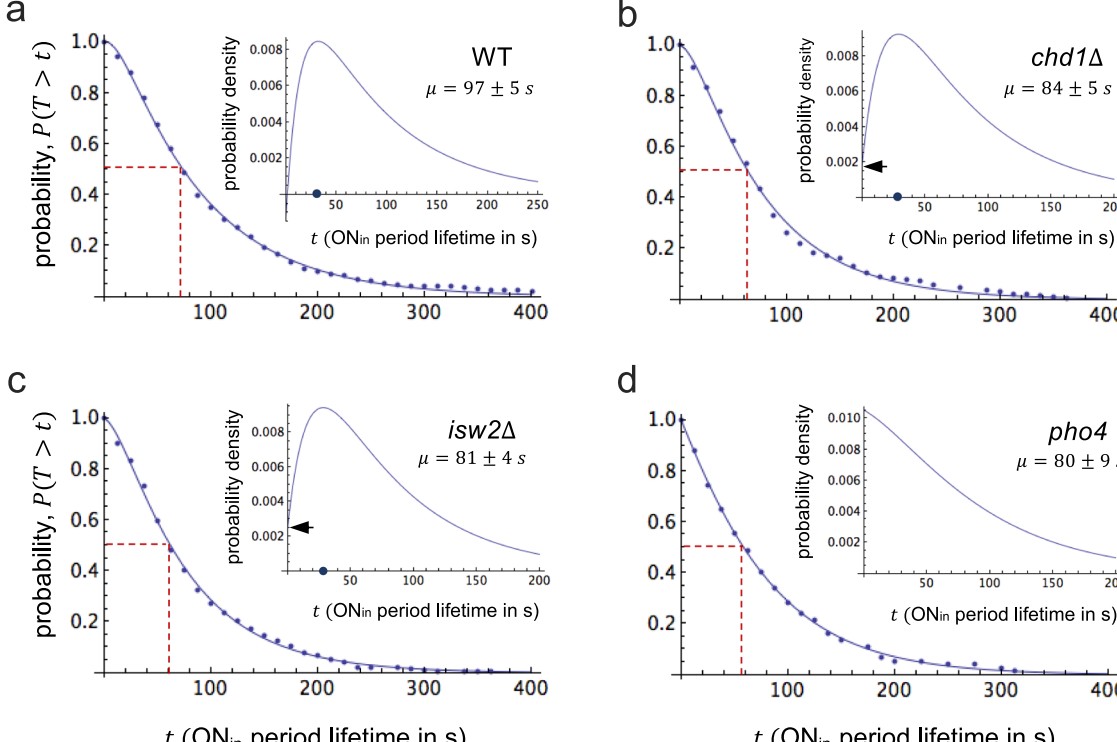

**Fig. 2 | Dwell-time densities are peaked − the observed process, therefore, is non-Markovian.** Survival curves of internal ON periods for **a** wild type, **b** *chd1Δ*, **c** *isw2Δ*, and **d** *pho4*. Measurements are indicated by blue dots and biexponential fits by blue curves. Inserts show the corresponding density function; $\mu$ is the mean period length (the average of 300 bootstrap replicates $\pm$ bootstrap standard deviation). The median of the distribution is highlighted by stippled red lines. Arrows point at the intercept of the density function with the ordinate for *chd1Δ* and *isw2Δ*. The average density at $t = 0.01$ across 300 bootstrap replicates (*cf.* "Methods") was $-0.001 \pm 0.003$ (wild type), $+0.002 \pm 0.0024$ (*chd1Δ*), $+0.0024 \pm 0.0025$ (*isw2Δ*), and $+0.01 \pm 0.004$ (*pho4*), $\pm$ the standard deviation across bootstrap replicas. Period lengths were obtained by CPD analysis (see "Methods") from 218 (wild type), 205 (*chd1Δ*), 291 (isw2Δ), and 158 (*pho4Δ*[75−90]) sample paths (cells), observed over a time period of 1250 s.

presumably by recruitment of ATP-dependent chromatin remodelers via its activation domain[17]. Loss of promoter nucleosomes allows Pho4 to access UASp2, a second, previously occluded enhancer[27,28]. At high media concentrations of orthophosphate, Pho4 is maintained in the cytoplasm; its target genes, therefore, remain inactive. However, in low phosphate or the absence of Pho80, a repressor of the phosphate signaling pathway, Pho4 accumulates in the nucleus, and *PHO5* is expressed[29]. All strains analyzed for this study were *pho80Δ*.

### Experimental setup

To observe transcription at single *PHO5* copies in vivo at the high temporal resolution, we built a multifocus fluorescence microscope (MFM), which allowed us to simultaneously acquire images at multiple focal planes, removing the spatiotemporal uncertainties associated with successive sampling of z-stacks[30]. *PHO5* transcripts were labeled by the insertion of 14 binding sequences for the RNA-binding coat protein of the PP7 phage into the 5′-untranslated region of *PHO5* (Fig. 1b)[31]. The coat protein was expressed as a fusion with the green fluorescent protein under control of the *RPS2* promoter (Fig. 1b). We acquired focal stacks − seven images spaced 500 *nm* apart along the optical axis − every 2.5*s* over a total period of 1250*s* (Fig. 1c).

To delineate periods of transcriptional activity (ON) and inactivity (OFF), we employed change point detection analysis (Fig. 1b, c)[32]. This delineation is part of the observational coarse-graining. We call periods whose length is known because they began and ended within the observational time window internal, and periods of unknown length because they either began or ended or began and ended outside the window of observation external. The length $T$ of ON and OFF periods is a random variable. We record the statistical distribution of $T$ in terms of its survival curve, which indicates the fraction (probability) of periods that are still alive at time $t$, $P(T > t)$, and the corresponding (probability) density of $T$, *i.e.*, the time-derivative of $P(T \leq t) = 1 - P(T > t)$.

### Dwell-time denisty functions are peaked

On the assumption that the observed dynamics can be modeled as a memory-free stochastic process (Markov process) that transitions between two microstates, ON and OFF (random telegraph model)[24], it is expected that survival curves fit an exponential decay function (*cf.* SI, Corollary 1). However, we found that the survival curve for ON periods was sigmoidal rather than exponential (Fig. 2a). Although the observed dynamics were non-Markovian (*i.e.*, endowed with memory), the survival curve was well fit by a biexponential function, suggesting that the observed dynamics may be modeled by a coarse-grained hidden Markov process with three microstates: two ON microstates, which are not distinguished by the observer and thus are combined into a single ON mesostate, and one OFF microstate (*cf.* SI, Theorem 1). (The limitation to one OFF microstate does not limit the generality of our conclusions, for the internal microstructure of the OFF state is irrelevant for the calculation of the ON dwell-time distribution, *cf.* SI). A biexponential survival curve implies a biexponential density function, $f(t) = c_1 e^{\lambda_1 t} + c_2 e^{\lambda_2 t}$, where $\lambda_1, \lambda_2 < 0$ and $c_1, c_2$ are constants − three of which ($\lambda_1, \lambda_2$ and $c_1$) may be independently chosen for fitting to the data (*cf.* "Methods"). Notably, the best biexponential fit suggested $f(0) = c_1 + c_2 = 0$ (*cf.* Fig. 2a). The density function, thus, must exhibit a local maximum, *i.e.*, the density is non-monotonic or peaked (*cf.* Fig. 2a, insert). For $f(0) = 0$, we say the density is maximally peaked.

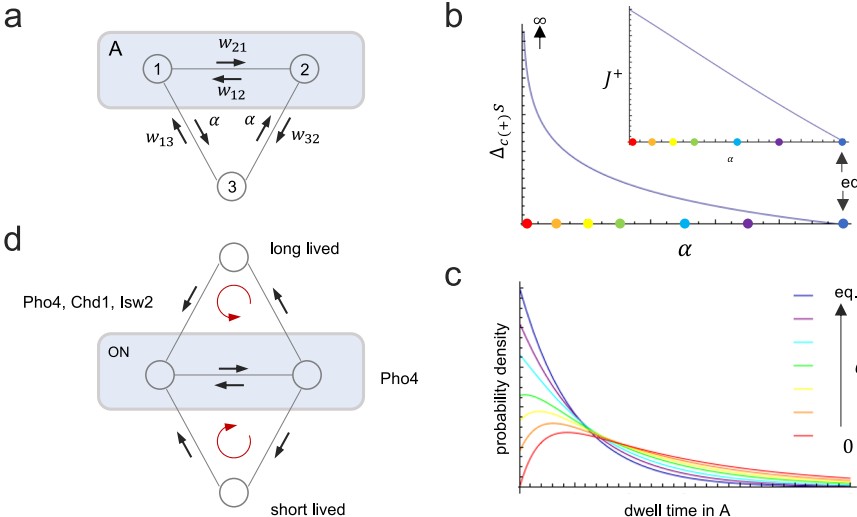

**Fig. 3 | Peaked dwell-time densities are indicative of entropy production.**
**a** Graph of the random process model for internal ON (or OFF) periods. Microstates are represented by circles. A blue rectangle indicates a mesostate due to coarse-graining. Labels on transition arrows are rate constants. **b** Steady-state entropy production per clockwise cycle, $\Delta_{c(+)}s$, and clockwise probability current, $J^+$, as a function of $\alpha$, ranging from $\alpha = 0$, maximally irreversible process, to $\alpha_{eq} = (w_{32}w_{21}w_{13}/w_{12})^{1/2}$, equilibrium process (which satisfies the cycle condition for detailed balance, $w_{32}w_{21}w_{13} = \alpha^2 w_{12}$). The rate of entropy production (increase in total entropy per unit time) is given by the Schnakenberg equation[56], which for a

cyclic steady-state process reduces to $\sigma = J^+ \Delta_{c(+)}s$[11] with $\Delta_{c(+)}s = 2k_B \ln\left(\alpha_{eq}/\alpha\right)$ and $J^+ = w_{13}p_3 - w_{31}p_1$, where $p_i$ is the steady state probability of state $i = 1,3$. Colored dots mark $\alpha$-values for which dwell-time densities in mesostate $A$ are plotted in the panel below: **c** Steady-state dwell-time densities in mesostate $A$ for selected $\alpha$-values. **d** Transition graph including states for both internal OFF periods (short-lived) and external OFF periods (long-lived). Labels on directed edges indicate the dependence of transitions on activator (Pho4) and/or remodelers (Isw2 and Chd1). The red arrow indicates the orientation of the net probability current in both the upper and lower cycles of the graph.

## The observed process is irreversible

The finding of a peaked dwell-time density has far-reaching topological and thermodynamics implications. First, dwell-time densities for processes on graphs with three states that are strongly connected − *i.e.*, every state is connected to any other by a string of directed edges − are always monotonic (*i.e.*, $c_1, c_2 \geq 0$), unless the reaction graph is cyclic (*cf.* SI, Theorem 2). The observed dynamics, therefore, must be modeled on a cyclic graph (*cf.* Fig. 3a). Second, dwell-time densities are always monotonic unless the process is irreversible − this is Tu's theorem[33] (*cf.* SI, Theorem 4). The observed process, therefore, was irreversible. Consistently, the only 3-state model that allows for a maximally peaked ON dwell-time density is the random clock model of Fig. 3a with $\alpha = 0$ (*cf.* SI, Theorem 3): the process strictly enters the ON state via one of its microstates and exits from the other; the reverse sequence is excluded. (This boundary case can be approached only asymptotically, for the definitive exclusion of reverse transitions violates the principle of microscopic reversibility and would require infinite entropy production, *cf.* Figure 3b.)

That a strongly peaked dwell-time density is indicative of a strongly irreversible process may be understood intuitively: The statistical preference for leaving the (observed) ON state *via* microstate 2 (*cf.* Fig. 3a) and not 1 (the microstate of entry into ON) delays exit from the ON state and thus reduces the probability of shorter ON periods.

In our 3-state clock model, a gradual decrease in the entropy production per cycle (increasing $\alpha$ in Fig. 3b), and thus a decrease in the statistical imbalance between forward and reverse paths, shifts the density at $t = 0$ from zero to larger values until it reaches a maximum in equilibrium (Fig. 3c). The density, thus, is gradually transformed from a maximally peaked function ($f(0) \leq f(t)$ for all $t > 0$) to a monotonic, non-peaked, function ($f(0) > f(t)$ for all $t > 0$; Fig. 3c). In equilibrium, when no entropy is produced, and forward and reverse direction of any stochastic path are equally probable[34], the dwell-time density is strictly monotonic, in accordance with Tu's theorem (Fig. 3b, c). The monotonicity of the dwell-time density constitutes a Hopfield barrier,

a bound that may be breached only at the expense of free energy, *i.e.*, away from equilibrium[35].

The distinction between sigmoidal and non-sigmoidal survival curves was subtle (*cf.* Supplementary Fig. 1). However, our conclusion of a peaked dwell-time density remained robust under perturbation: $c_1$ remained negative for all densities generated by bootstrapping (random sampling with replacement); only a small number of densities (< 2%) had peaks close enough to $t = 0$ such that the density appeared monotonic; the remaining densities (>98%) were manifestly peaked (Supplementary Fig. 2a).

With $f(0) = 0$, the best-fit biexponential density for the internal OFF period length was likewise maximally peaked (Supplementary Fig. 3), suggesting that both entry into and exit from the OFF state were strongly irreversible − as implied by our analysis of ON lengths − and therefore represented not forward and reverse path of the same reaction but the forward paths of two different reactions. This conclusion, evidently, depends on the ability to decompose ON or OFF into multiple microstates. The existence of two rather than one microstate was less evident for the OFF than the ON state: although $c_1$ was negative for 98% of bootstrap densities, nearly 50% of biexponential fits peaked close to $t = 0$ and thus were difficult to distinguish from the exponential function.

## The regulatory process dissipates free energy

The observed process is a complicated convolution of multiple processes: the regulatory process for transcription, subordinate molecular processes, such as polymerization and nuclear export of the RNA, the RNA-binding dynamics of the fluorescent protein, and the physical processes within the instruments of observation − the microscope, CCD camera, laser, etc. It may be asked, therefore, whether the irreversibility of the observed process was due to energy-consuming processes other than the regulatory process mechanism for transcription.

To test this latter hypothesis, we deleted parts of the activation domain of Pho4, the central, gene-specific component of the

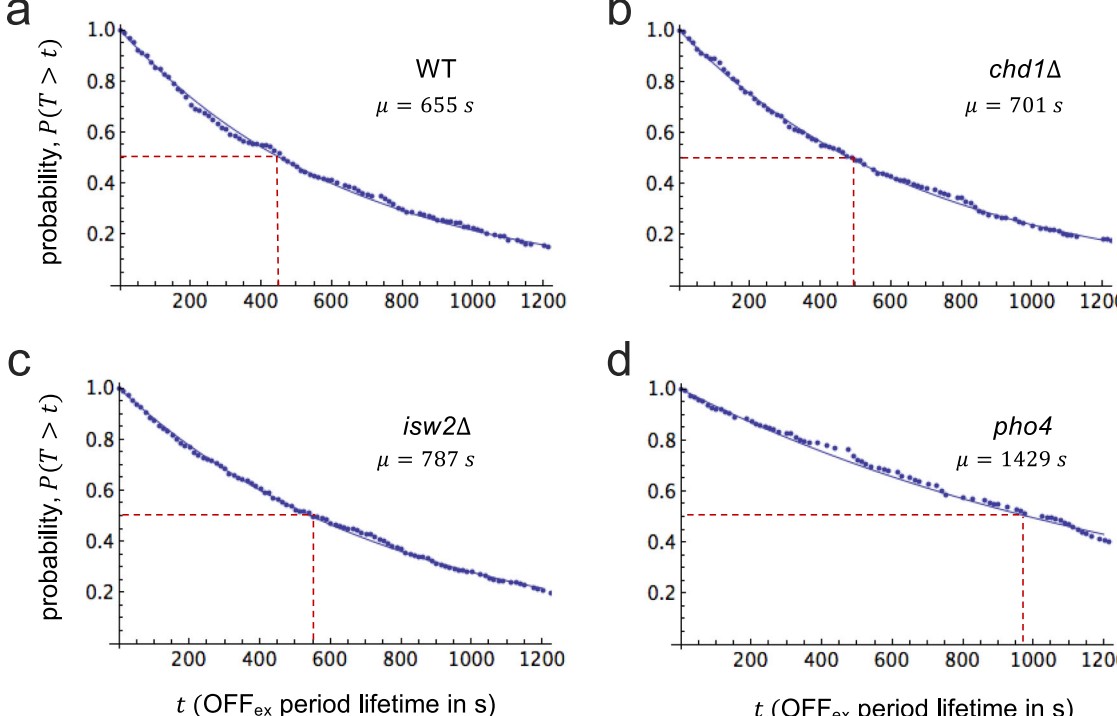

**Fig. 4 | The length of external OFF periods is controlled by both the Pho4 activator and chromatin remodelers. a** Survival curves of external OFF periods for wild type, **b** *chd1Δ*, **c** *isw2Δ*, and **d** *pho4*. Measurements were fit by a single exponential, *i.e.*, $P(T>t) = e^{-\lambda t}$ (blue curves). The corresponding densities, $f(t) = \lambda e^{-\lambda t}$ (not shown), are strictly monotone (non-peaked). The median of the distribution is highlighted by stippled red lines; $\mu$ is the mean period length (the average of 300 bootstrap replicates).

regulatory mechanism. The deletion reduced *PHO5* expression by about 80% and markedly increased promoter nucleosome occupancy relative to wild type (*cf.* Supplementary Fig. 4)[14,36]. According to the nucleosomal proofreading hypothesis, the activator mutation disrupts the energy supply line for the regulatory process by weakening the activator's ability to recruit ATP-dependent chromatin remodelers to the promoter; the process relaxes, therefore, into a new steady state closer to equilibrium, where nucleosome removal consumes, and reformation dissipates, less free energy per cycle, on average. We found that the activator mutation transformed the maximally peaked into a non-peaked distribution (Fig. 2d), as expected for the transformation of a strongly irreversible into a weakly irreversible regulatory process (*cf.* Fig. 3c and b). Alternative explanations must invoke unexpected pleiotropic effects − *e.g.*, that *pho4* mutations somehow affect the energetics of RNA polymerization.

The ON dwell-time densities for the activator mutant and wild type were clearly distinct: While the densities of 90% of bootstrap replicates for the mutant were monotonic or weakly peaked, only 1.3% of bootstrap densities for the wild type were monotonic, the remainder were strongly or maximally peaked (Supplementary Fig. 2a, b).

**Atp-Dependent nucleosome removal drives the regulatory process away from equilibrium**
Loss of the ATP-dependent chromatin remodelers Chd1 and Isw2 reduced *Pho5* activity by about 30% and 50%, respectively, as shown by phosphatase activity measurements. Electron microscopy analysis of *PHO5* molecules isolated from *chd1Δ* cells showed a concomitant increase in promoter nucleosome occupancy relative to wild type (Supplementary Fig. 4). For both remodeler mutants, the density for the internal ON period length was again peaked. Fewer than 1% of bootstrap densities were apparently monotonic. However, for both remodeler mutants, we found $f(0) = c_1 + c_2 > 0$ (for the best-

biexponential fit) and not $c_1 + c_2 = 0$; the densities were strongly but not maximally peaked, corresponding to a process that dissipated less free energy (produced less entropy) per cycle than the wild type (*cf.* Fig. 3c and d). That nucleosome occupancy increased with decreasing entropy production was as expected if nucleosome removal, and not reformation, drives the regulatory process away from equilibrium, *i.e.*, the probability current flowed in the correct direction for proofreading (clockwise in Fig. 1a). We note however that values for $c_1 + c_2$ varied widely among bootstrap replicates (*cf.* legend to Fig. 2).

**Activator and chromatin remodelers control the length of external but not internal off periods**
All mutations analyzed had little effect on the average length of internal ON periods and, surprisingly, also internal OFF periods (*cf.* Figs. 2, 4, and Supplementary Fig. 3): ON periods became shorter and OFF periods longer by about 15% in all mutants compared to wild type. The main effect of activator and remodeler mutations became evident upon analysis of *external* OFF periods. The length of external OFF periods, $T$, was exponentially distributed (Fig. 4). $T$ could be modeled, therefore, as a sojourn of the process in a single, additional microstate (Fig. 3d). Although $T$ could not be known for individual periods, the lack-of-memory property of the exponential distribution nevertheless allowed us to determine mean period lengths[37]. On average, external OFF periods were significantly longer than internal OFF periods; the process either left the ON state for a short-lived OFF state or a long-lived OFF state (Fig. 3d). Since the lifetime of long-lived OFF states was also long relative to the length of the observational time window, the process seldom returned to the ON state within the observational time window after leaving the ON state for the long-lived OFF state. Exit from the long-lived OFF microstate was promoted by both activator and chromatin remodelers, for external OFF periods markedly

increased in length in all three mutants compared to the wild type in order of decreasing *PHO5* expression (*cf.* Fig. 4).

## The regulatory mechanism controls both frequency and length of burst runs

Individual ON periods, in all likelihood, corresponded to bursts of transcription with multiple nascent transcripts. Genes with fewer than three transcripts may not have been observable (Supplementary Fig. 5). Except for the activator mutant, ON periods were more probably succeeded by short-lived rather than long-lived OFF periods. Transcription events, thus, were two-fold clustered: into bursts (ON periods) and bursts of bursts (burst runs). The sequence-specific activator and chromatin remodelers controlled the frequency of burst runs alone, not the frequency of bursting within runs, the frequency of transcription initiation within bursts, or the rate of transcription itself. Furthermore, the probability that an (internal) ON period was followed by a long-lived and not short-lived OFF period increased nearly two-fold in the *pho4* mutant, from 0.39 (wild type) to 0.65, indicating that the activator controlled the length of burst runs. The activator, thus, exerted control over the rate of transcription initiation in more than one way. In contrast, the loss of Chd1 and Isw2 had little effect on the length of burst runs. The finding that chromatin remodelers affected a subset of steps affected by the activator and not others is in keeping with the notion that activators recruit chromatin remodelers to genes[18,19], but also control subsequent steps toward transcription by recruitment of additional factors, *e.g.*, Mediator[38,39]. Both assumptions are critical for the theory of nucleosomal proofreading of activator identity[10].

## Discussion

The main finding of our analysis is that the regulatory process of transcription initiation operates away from equilibrium. Our analysis detected a telltale of irreversibility: peaked dwell-time densities for observational states[33]. Peaked dwell-time densities have been reported earlier[40,41]; however, the topological and thermodynamic implications went unnoticed, and the question of whether the regulatory process was irreversible was, therefore, not addressed.

A mutation in the central component of the regulatory mechanism for transcription transformed the observed process from a strongly irreversible into a weakly irreversible process, suggesting that the observed process was irreversible not because RNA synthesis and the instruments of observation dissipate free energy (which they do) but because the regulatory process was irreversible.

The observed ON-OFF dynamics are the result of extensive coarse-graining, which may easily transform an irreversible into a reversible process (*cf.* Introduction). It may be asked, therefore, whether the reverse is possible too, *i.e.*, whether coarse-graining may convert a reversible into an irreversible process? The answer, arguably, is no, for it can be proved that the entropy production of any coarse-grained version of a stochastic process provides a lower bound on the true entropy production (*cf.* SI, Theorem 5)[22,42].

When fit to the data, the hidden Markov process provides a model for the observed dynamics, not the underlying molecular dynamics. Additional microstates were invoked to imbed the observed non-Markovian process into a Markov process. The identification of model states with particular microstates of the underlying molecular process is, therefore, entirely speculative. The OFF state evidently encompasses numerous microstates, and so does the ON state − *e.g.*, *PHO5* transcription may be driven by alternative activator-enhancer configurations. It is not evident how to interpret (in this way) the finding that the process when out of equilibrium, visits ON microstates in a defined order.

Our analysis was undertaken to test a critical prediction of the kinetic proofreading hypothesis for activator identity: that the regulatory process for transcription initiation is irreversible. The

hypothesis passed its test. The alternative conclusion − that the regulatory process is reversible − would have falsified the proofreading hypothesis in any form. Instead, our findings (if accepted) refute the equilibrium assumption and suggest that the regulatory process is driven away from equilibrium. The driving, it appears, is at least partially due to ATP-dependent chromatin remodelers, as posited by the nucleosomal proofreading hypothesis.

Our results have no direct bearing on the question of whether the regulatory dissipation of free energy improves the specificity of transcriptional regulation. Other problems of information processing may be solved away from equilibrium[3,35,43]. Also, the possibility of free energy expenditure for no physiological purpose whatsoever is not excluded. However, if the nucleosome dynamics of transcriptionally induced, but not uninduced, promoters are out of equilibrium, as suggested by our analysis, the conclusion that the structural dynamics of promoter chromatin improve activator specificity follows with little effort[7,10].

The problem of regulatory specificity may not be equally acute for all genes. Housekeeping genes, genes that are active at all times, may rely on promiscuous activator control or, perhaps, no activators at all. Such genes have no need for activator proofreading. Their promoters may be, therefore, almost always nucleosome-free. Whether such promoters exist is currently unknown. Constitutive lack of promoter nucleosomes was implicitly invoked as part of an explanation for sub-Poissonian transcription noise[43], i.e., noise below the expectation for a random birth-and-death (Poisson) process[35]. (In contrast, the noise of *PHO5* transcription markedly exceeded the expectation of a Poisson process.) We note that sub-Poissonian RNA noise also requires free energy expenditure − arguably to drive promoter state transitions and thus narrow the distribution of interarrival times between transcription events[43].

## Methods
### Strains

All strains for live cell fluorescent microscopy were derived from yM8.14, which contains sequence insertions upstream and downstream of the *PHO5* gene for enzymatic release of the *PHO5* gene from its chromosomal locus and lacks the *PHO80* gene, which was replaced with *HIS3*. We inserted the PP7 binding sequence clusters into the *PHO5* locus via homologous recombination. To this end, the cluster, along with a selection marker (KanMX) flanked by loxP recombination sequences, was PCR amplified from plasmid pTL31 using primers p446 (5′-CTTCATCTCTCTCATGAGAATAAGAACAACAACAAATAGAGCAA GCAAATTCGAGATTACCAcaaagtgggagcgaggagatcc-3′) and p447 (5′-AATGGTACCTGCATTGGCCAAAGAAGCGGCT AAAATTGAATAAACAA-CAGATTTAAACATgcataggccactagtggatctg-3′) whose 5′-ends are homologous to the *PHO5* 5′ UTR. This insertion yielded strain yM249.1. Subsequent removal of the KanMX marker was accomplished by transformation with pSH47, which bears the P recombinase gene, and induction of recombinase expression. This gave strain yM255.3. Finally, we inserted the gene for the PCP (PP7 coat protein)-GFP fusion together with the *URA3* gene of *Candida albicans* into the *ura3* locus of yM255.3. To this end, we transformed yM255.3 with pTL205 (a derivative of pSIVURA3[44]) digested with PacI and selected for uracil prototrophy. The strain thus generated, yRS102, is called here wild type for its possession of wild type copies of *CHD1*, *ISW2*, and *PHO4*. Strains yRS99 (chd1Δ : KanMX) and yRS118 (isw2Δ : KanMX) were obtained by transformation of yM255.3 with pCM123 digested with XbaI and KpnI and pCM122 digested with XbaI and KpnI, respectively, and subsequent insertion of the PCP-GFP fusion gene into the *ura3* locus, as described above. The activator mutant strain, yRS120, was derived from yM255.3 by first replacing *PHO4* with *URA3 via* homologous recombination using pCM4.5 digested with BamHI and SalI. The resulting strain, yRS103, was transformed with pCM64.2 cut with BamHI and SalI to replace *URA3* with *pho4*Δ[75−90]. This yielded

yRS113 upon selection for resistance to 5-fluoroorotic acid (5-FOA). We finally inserted the gene for the PCP-GFP fusion into the *ura3* locus of yRS113, as described above. Cells were transformed using the lithium acetate method[45].

## Live-cell imaging

Yeast Cells were grown in 50 ml of liquid culture to $3 \cdot 5 \times 10^7$ cells/ml. Cells were then concentrated and spotted onto an agar patch on a #1.5 coverslip, as described elsewhere[46]. For coverslip preparation we used the mold described by ref. 47. Prior to imaging, cells were then incubated at 30 $\mathcal{C}$ for 30 minutes. A stage top incubator (OkoLab) was used to maintain a constant temperature of 30 $\mathcal{C}$ during imaging. MFM images were recorded with on an EMCCD camera (Andor Ixon Ultra). Images were taken every 2.5 second over 20 minutes with an exposure time of 250 *ms* and a laser power of 30% (Lambda XL). Excitation and emission paths were filtered using a Semrock GFP-30LP-B Brightline filter set. The MFM was assembled as described[30,48] and appended to the light path of an existing inverted widefield microscopy chassis (Nikon Eclipse TI), which maintained a $60 \times$ objective (plano Apo 60X/1.4 oil immersion).

MFM raw data were analyzed in four steps: (*i*) Identification of transcription sites. (*ii*) Assignment of transcription sites to nuclei. (*iii*) Tracking of each transcription site over time. (*iv*) Quantification of signal intensity at transcription site.

To identify transcription sites, we first calculated a maximum projection of the z-stack for each time point; the maximum projection minimizes fluctuations in puncta intensity due to RNA movement in the *z*-direction (*i.e.*, along the optical axis) during imaging. We then applied a band-pass filter to maximum projections with bandwidth to match the width of the point spread function of the microscope (approximately 1.5 pixels); the band-pass filter reduces the rate of false positives due to background nuclear fluorescence caused by cytoplasmic RNA and unbound coat protein[49,50]. To find local maxima (*i.e.*, candidate transcription sites), we applied a search window of 4 pixels and a threshold of five standard deviations above the mean filtered image signal. We assigned transcription sites to nuclei, using the background nuclear fluorescence to determine nuclear boundaries. We always identified the brightest site within a nuclear boundary as the site of transcription. Time series were generated by following individual transcription sites over time. When no candidate transcription site could be detected within a nucleus at a given time point, the location of the previously identified transcription site was used instead. In the case of no previous transcription site, we used the location of the brightest nuclear pixel instead. Finally, to quantify the signal intensity of transcription sites, we used a Gaussian mask algorithm[49,50].

We corrected for photobleaching by detrending time series; a trend was inferred by regression analysis of the average time series[49,51]. Variation in nuclear intensity creates variations in background intensity. To correct for offsets of time series against each other along the *y*-axis (signal intensity), we subtracted from each time point the mode of the estimated probability function (kernel density estimate) of signal intensity of each time series. The mode was used because, unlike mean and median, it correctly reflects the background on the assumption that the most likely number of nascent transcripts is zero − an assumption borne out by our observations.

We assumed that sample paths (*i.e.*, fluorescence signal traces) were step functions subject to Gaussian noise. Steps, time intervals of signal that appeared to be drawn from the same Gaussian distribution, were determined by change point detection (CPD) analysis[32,49]; we used a window-based search method together with a Gaussian cost function in the ruptures python package[31]. We observed essentially two types of steps: low-noise steps at lower signal (OFF steps) and high-noise steps at higher signal (ON steps). CPD hyperparameters (*e.g.*, window size) were selected to reduce the number of sequential low-

noise steps on the assumption that the background signal, the signal during OFF periods, is step-free. In contradistinction, the transcription signal, the signal during ON periods, may encompass multiple steps corresponding to different numbers of the nascent transcript. For wild type we analyzed 218 sample paths (cells); for *chd1Δ*, 205 cells; for isw2Δ, 291 cells, and for *pho4Δ[75−90]*, 158 cells.

## Model fitting

The survival-function model, $P(T > t) = -ae^{-\lambda_1 t} + (1+a)e^{-\lambda_2 t}$, was fit to the experimental data using the FindFit function in Mathematica (Wolfram) with search parameters $a, \lambda_1, \lambda_2$, and variable $t$[49]. The corresponding density function is given by $f(t) = c_1 e^{-\lambda_1 t} + c_2 e^{-\lambda_2 t}$ with $c_1 = -a\lambda_1$ and $c_2 = (1+a)\lambda_2$. The search algorithm was left unconstrained, which allowed for negative densities. When FindFit was constrained by $-a\lambda_1 + (1+a)\lambda_2 \geq 0$ to densities $0 \geq$ at $t = 0$, the parameters satisfied $c_1 + c_2 = 0$ exactly for internal ON periods in the wild type. The constrained and unconstrained fits were essentially identical except for $t = 0$ and time points very close to it. Bootstrapping analyses were performed with an unconstrained Mathematica FindFit function[49]. For some bootstrap replicas, we found $\lambda_1 \gg \lambda_2$. The corresponding density functions, therefore, assumed either large negative or large positive values at $t = 0$. In these cases, the density function was virtually discontinuous at $t = 0$ and indistinguishable from the exponential density, $f(t) = \lambda_2 e^{-\lambda_2 t}$.

## Single-molecule fish

Yeast cells were grown in liquid culture ($50 ml$) to mid-log-phase, crosslinked with formaldehyde, lysed with lyticase, and adhered to poly-L-lysine coated coverslips, as previously described[52,53]. Coverslips were hybridized for 5 hours at 37 °C, with 2.5 *nM* of fluorescently labeled antisense-DNA probe against PCP binding sequences. Probes were labeled with Quasar 570 and Quasar 670 (Biosearch Technologies). For microscopy, coverslips were mounted onto glass slides with mounting media containing DAPI (4′, 6-diamidino-2-phenylindole; ProLong Gold, Life Technologies). Cytoplasmic and nascent RNA were identified by using background fluorescence and DAPI staining to estimate cellular and nuclear boundaries, respectively. The number of nascent RNAs was calculated by normalizing the transcription site intensity (nuclear punctum) with the average intensity of cytoplasmic transcripts. All analysis was accomplished using previously published code[54].

## Pho5 Isolation and Em analysis

Gene isolation and psoralen-EM analysis were performed as previously described[55]. Molecules were analyzed with custom-made programs written in Python. At least 200 *PHO5* molecules were analyzed for each strain[14].

## Reporting summary

Further information on research design is available in the Nature Portfolio Reporting Summary linked to this article.

## Data availability

The MFM data of this study are available at https://datadryad.org/stash/share/p-nEcqmpD7aj4RUSfzuVmR2kZ_iBEvh1pwf8OqBeO1Y.

## Code availability

All computer codes used are available at https://doi.org/10.5281/zenodo.12208276.

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

## Acknowledgements

H.B. and S.A. gratefully acknowledge support from the National Science Foundation (awards #2111763 and #1828636, respectively). We thank John Field and Heta Patel for their critical comments on the manuscript. H.B. is indebted to Jeremey Gunawardena for stimulating discussions.

## Author contributions

R.S. generated strains, conducted MFM, single molecule-FISH, analyzed the primary data, and wrote Python programs for data analysis. S.A. built the microscope for MFM together with R.S. C.R.B. and M.D. isolated *PHO5* molecules and analyzed single molecules by psoralen-crosslinking and electron microscopy. T.L. and D.R.L. provided software and training to R.S. H.B. conceived the project, analyzed dwell time distributions, and wrote the manuscript.

## Competing interests

The authors declare no competing interests.
