## [Peer Review File · Nature Communications]

Single Gene Analysis in Yeast Suggests Nonequilibrium Regulatory Dynamics for TranscriptionREVIEWER COMMENTS

Reviewer #1 (Remarks to the Author):

Summary: The goal of this study is to demonstrate the non-equilibrium nature of transcription initiation in yeast cells using live-cell experiments and theory. The authors use nascent transcription experiments to reconstruct time periods where the considered gene was in transcriptionally active vs. inactive states. Based on the resulting waiting-time distributions, the authors derive conditions which allow them to conclude whether the observed dynamics is irreversible and to what extent. Overall I found the paper interesting and the results are compelling. That transcription regulation is a non-equilibrium process is perhaps not too surprising but there are few studies providing evidence for this. The paper is within the scope of nature communications but I have several technical questions and criticisms, which I believe should be addressed.

Major comments:

-An important assumption underlying the author's framework is that the underlying "microstates" exhibit Markovian dynamics, such that the waiting-time distributions of the "mesostates" are functions of exponentials. Or in other words, with the approach proposed by the authors, one always ends up with a description of the underlying dynamics in terms of Markov states, similar to the idea of phase type distributions. Since the conclusions about irreversibility strongly depend on this assumption, I feel it should be discussed further. For instance, is it possible that there are alternative realizations of microstates with non-Markovian dynamics that are consistent with the data that have a different degree of irreversibility?

-The detection of on / off times depends on a change point detection algorithm. First, given the importance of this approach to the overall study, I felt that the description of it was quite superficial. Second, it is unclear how such statistical reconstruction of the on/off times affects the inferred waiting time distribution. Depending on the underlying system dynamics and measurement noise, one can imagine that the resulting waiting times may look quite different from the "true" ones, for instance if two bursts happen one after another (in case two on periods may be merged to a single one). A systematic numerical analysis on simulated ground-truth data would be helpful to obtain a sense of the robustness of the reported results.

-This is related to the previous comment. The authors infer on / off times from PP7 signal, which is a proxy for nascent RNA. However, this signal should be convolved with elongation as well as transcripts leaving the gene once fully transcribed. How will this affect the inferred on/off periods? Another way to ask the same question is: how exactly do the authors define the on-state? In the manuscript I found that the on state corresponds to "transcriptional activity", which could be defined as a state where there is any significant PP7 signal, but on the other hand, the authors draw conclusions about the "regulatory process of transcription initiation", for which such definition based on PP7 signal appears less clear.

-I could not follow the discussion around internal vs. external off periods. Why would one expect external periods to be any different than internal ones? As far as I understood, the gene is constitutively expressed so I would expect the waiting-time statistics to be more or less constant over time. I agree with the statement that exponential waiting-times do not have to be observed entirely in order to estimate their mean lifetime, but why would the mean lifetime change depending on when one observes? I must have missed something, but I would be grateful if the authors could clarify this point.

-Is it really true that the regulatory processes controlling transcription are "generally assumed to

operate reversibly"? It is true that (quasi-) equilibrium arguments are frequently employed to simplify models of transcription regulation, but I am not sure whether the overall consensus is that transcription regulation is at or close to equilibrium.

Minor comments:

-Figure 3 lacks a description of panel (D)

Reviewer #2 (Remarks to the Author):

Here the authors expand on the transcription model proposed by Boger in his 2022 review of kinetic proofreading being an important mechanism in eukaryotic transcription. This is an important topic and can help explain how cells balance reaction speed with specificity and that the observed specificity of the transcription process is difficult to explain by the specificities of the individual components (e.g., DNA binding proteins...). Although the ideas presented here seem important, a major problem for me is that the manuscript is not accessible to a general audience, and it was very difficult for me to understand the logic and details of the author's evidence and conclusions. In my opinion, in order for this to be considered for a non-specialist journal, the manuscript needs a major rewrite to better explain the logic of the study, the results and their implications. Examples of these issues (not a list of all of them) are given below:

Lines 41-45: in the introduction, the requirement for a repressor in kinetic proofreading seems to come out of nowhere – is a repressor necessary? I think that the introduction could be vastly improved to make this understandable for a general audience.

Line 84: this points to fig 1, panels c-j but these panels don't exist. It looks like these labels were removed in the current figure. Also, Fig 1 says that panel a has a green triangle, but it's a grey triangle.

Line 98: what is an on/off mesostate?

Figure 1 legend says "box contains 100 binary time series" but it looks like the lower right box contains many fewer time series.

Fig 3B: upper and lower panes are confusing – all the points are zero on the Y-axis so where did the blue lines come from?

In the formula on line 94, c_1 , c_2 and $\lambda_{1,2}$ are not defined and explained as far as I can tell. This is a major problem as $c_{1,2}$ are brought up multiple times in the manuscript.

Line 100 – c_i is not defined; c_j not defined on line 105.

I don't understand how the data support burst runs (line 190 and following)

Line 211: the authors don't explain well how their data strongly supports a kinetic proofreading mechanism – the main point of their paper.

I can't follow the logic in the paragraph starting on line 103.

Line 166: what is a pho5 chromatin ring?

Reviewer #3 (Remarks to the Author):

Overall, this is a nice work. By using the Tu's theorem (ref. 31) on general properties of dwell time distributions of any equilibrium systems, the measurements of dwell time distributions of the gene expression dynamics reported in this study convincingly show that gene transcription is a non-equilibrium process. The proposed three-state model also gives a set of interpretable predictions to which the measurements can be compared with to further verify the model. There are, however, a few comments and questions we would like the authors to address/clarify before we can fully support publication of this work:

- 1) The observed peak in dwell time distribution indicates that the underlying gene expression dynamics operates out of equilibrium with energy dissipation. The illustrative Fig. 1A seems to suggest that the source of energy is ATP hydrolysis. The natural question is how the system uses the ATP hydrolysis energy to drive the reaction cycle as shown in Fig. 3D, i.e., what is the molecular mechanism for the dissipative gene expression process?
- 2) A related question to the previous one is what are the functional benefits to carry out such a dissipative gene expression process. For example, in other recent studies of nonequilibrium biological systems such as sensory adaptation (Lan et al, Nature Physics 8, 422–428, 2012) and synchronization of molecular clocks (Zhang et al, Nature Physics 16, 95-100, 2020), a strong case has been made in terms of a general tradeoff relationship between the energetic cost of the systems and their performance benefits. The question here is whether the nonequilibrium gene expression mechanism, which costs chemical energy, has any functional benefit to the system?
- 3) Lines 137-144: The authors say the OFF period distribution is also strongly peaked suggesting strongly irreversible entry and exit. But then say 50% of bootstrap densities reduced to exponentials and use a single microstate to model the OFF state. I found this contradictory: it seems like the measurements do not consistently support a strongly irreversible process for the OFF state.
- 4) In the model (Fig. 3 B-C) the density becomes monotonic at a particular entropy production level. Does this level have any biological or theoretical significance? Can the model be used to (at least roughly) bound the entropy production level in the real system?
- 5) Fig. 1A provides useful intuition, but the finding in this paper is that the ON state has the irreversible multi-state structure. Is there a similar biological interpretation of the two ON microstates?
- 6) The difference between "internal" and "external" OFF timescales does indicate the presence of both short- and long-lived OFF states. But the criteria for separating these (whether the OFF period intersects the boundary of the observation window) can conflate the two: shorter dwell times in the long-lived state may be classified as "internal" and longer dwell times (or that start near the end of the observation) in the short-lived state may be classified as "external". Can this effect bias or otherwise influence the measured statistics?
- 7) Theorem 1 in the SI assumes the eigenvalues of the restricted transition matrix $W_{\{AA\}}$ are real. But this is not generally true for non-equilibrium systems. The exponential decomposition still holds, but for complex eigenvalues with negative real part.

Minor issues:

Fig. 2 & Extended Data Fig. 3: might be useful to mark the peak of the distribution instead of (or in addition to) the median.

Caption describing Fig. 3D missing, it is unclear what the mutant labels in this panel refer to.

Fig 4 caption is wrong: should say "external OFF periods"

References missing in SI?

SI Footnote 1 formatting typo

SI pg. 2: "diagonal elements of W are contained in the diagonal matrices $W_{\{AA\}}$ and $W_{\{BB\}}$," these matrices are not diagonal

Nonequilibrium Dynamics of Transcription Initiation: Evidence from Single Gene Analysis in Yeast

POINT BY POINT RESPONSE TO REVIEWERS

We thank reviewers for their careful reading of the manuscript and stimulating comments. In response, we have made numerous changes to the manuscript. Especially the middle section of the paper, pages 4-5, was mostly rewritten. We also have amended the Supplemental Information by two proofs with the aim to make our argument self-contained. We provide proofs of "Tu's theorem" (Section IV, Theorem 4) and the statement that any coarse-grained version of the true process provides a lower bound on the true entropy production (Section V, Theorem 5), for both statements play critical roles in our analysis.

The reviewers' comments are addressed below point-by-point; reviewer comment are in Arial, followed by our response in Times New Roman.

Reviewer #1

Major comments:

An important assumption underlying the author's framework is that the underlying "microstates" exhibit Markovian dynamics, such that the waiting-time distributions of the "mesostates" are functions of exponentials. Or in other words, with the approach proposed by the authors, one always ends up with a description of the underlying dynamics in terms of Markov states, similar to the idea of phase type distributions. Since the conclusions about irreversibility strongly depend on this assumption, I feel it should be discussed further. For instance, is it possible that there are alternative realizations of microstates with non-Markovian dynamics that are consistent with the data that have a different degree of irreversibility?

Our argument does not rely on the assumption that the underlying molecular process was Markovian, only that the *observed* process can be modeled by a coarse-grained Markovian process, in good agreement with the data. That the underlying molecular process is irreversible if the observed process is irreversible does not depend on the Markov assumption. We agree with the reviewer that this argument warrants further elaboration. To this end, we have amended the Discussion (third to last paragraph, lines 263-271) and we have included in the Supplementary Information a proof of Theorem 5 (*cf.* above).

The detection of on / off times depends on a change point detection algorithm. First, given the importance of this approach to the overall study, I felt that the description of it was quite superficial. Second, it is unclear how such statistical reconstruction of the

on/off times affects the inferred waiting time distribution. Depending on the underlying system dynamics and measurement noise, one can imagine that the resulting waiting times may look quite different from the "true" ones, for instance if two bursts happen one after another (in case two on periods may be merged to a single one). A systematic numerical analysis on simulated ground-truth data would be helpful to obtain a sense of the robustness of the reported results.

The change point detection algorithm is part of the observational coarse-graining. The reviewer is correct, different coarse-graining will generate different waiting time distributions. However, the entropy production of *any* coarse-grained version of the original process provides a lower bound on the true entropy production (*cf.* Theorem 5).

This is related to the previous comment. The authors infer on / off times from PP7 signal, which is a proxy for nascent RNA. However, this signal should be convolved with elongation as well as transcripts leaving the gene once fully transcribed. How will this affect the inferred on/off periods? Another way to ask the same question is: how exactly do the authors define the on-state? In the manuscript I found that the on state corresponds to "transcriptional activity", which could be defined as a state where there is any significant PP7 signal, but on the other hand, the authors draw conclusions about the "regulatory process of transcription initiation", for which such definition based on PP7 signal appears less clear.

The ON state is defined operationally, *via* change point detection analysis. We have rewritten major parts of the Results section to more clearly draw a distinction between the underlying molecular process and the coarse-grained observational process. The reviewer is correct, that the observed process is a convolution of the regulatory process for transcription initiation and the process of transcription. (*cf.* Lines 183-184.) Our conclusion that the regulatory process is irreversible is based on the (uncontroversial) assumption that Pho4 does not affect the thermodynamics of RNA polymerization — if it were, *pho4Δ* cells would be dead. Yet, the activator mutation transformed a peaked dwell time density into non-peaked density, as expected for the transition from a strongly irreversible process to a nearly reversible process. If transcription remained irreversible, then the regulatory process must have come closer to equilibrium.

I could not follow the discussion around internal vs. external off periods. Why would one expect external periods to be any different than internal ones? As far as I understood, the gene is constitutively expressed so I would expect the waiting-time statistics to be more or less constant over time. I agree with the statement that exponential waiting-times do not have to be observed entirely in order to estimate their mean lifetime, but why would the mean lifetime change depending on when one observes? I must have missed something, but I would be grateful if the authors could clarify this point.

That external OFF periods were significantly longer than internal OFF periods is an experimental result. We agree, this is surprising. Our explanation is this: There are two types of OFF states, stable and less stable, into which the process may jump after exiting the ON state. Since the lifetime of the stable OFF state was long relative to the length of the observational time window,

the process generally did not return to the ON state within the observational time window if exiting into the stable, rather than unstable, OFF state (*cf.* Lines 227-232).

Is it really true that the regulatory processes controlling transcription are "generally assumed to operate reversibly"? It is true that (quasi-) equilibrium arguments are frequently employed to simplify models of transcription regulation, but I am not sure whether the overall consensus is that transcription regulation is at or close to equilibrium.

By positing non-cyclic transition graphs, most random process models of transcriptional regulation, including the standard random telegraph model, tacitly assume equilibrium (*cf.* references 3-5). We presume that earlier equilibrium models were suggested on the assumption that they might be correct. Our analysis argues otherwise.

Minor comments:

Figure 3 lacks a description of panel (D)

The lacking description has been added.

Reviewer #2 (Remarks to the Author):

Although the ideas presented here seem important, a major problem for me is that the manuscript is not accessible to a general audience, and it was very difficult for me to understand the logic and details of the author's evidence and conclusions.

The manuscript was largely revised with the aim to increase comprehensibility.

Lines 41-45: in the introduction, the requirement for a repressor in kinetic proofreading seems to come out of nowhere – is a repressor necessary? I think that the introduction could be vastly improved to make this understandable for a general audience.

For kinetic proofreading, any free-energy dissipating delay mechanism might do; the requirement for the removal of a repressor provides such a delay step; and for the kinetic proofreading of activator identity a repressor (the nucleosome) was explicitly invoked (*cf.* Shelansky and Boeger, 2020). We agree with the reviewer that our introduction by no means does the matter justice. We have made several edits to improve our exposition. However, any satisfactory account requires more effort and space than seems suitable for the purpose of this study, whose aim was to determine whether the regulatory process for transcription is reversible or irreversible. The aim of our Introduction is merely to show that this is an interesting question, for a theory exist (*viz.* nucleosomal proofreading of activator identity) which would be refuted if the process were reversible.

Line 84: this points to fig 1, panels c-j but these panels don't exist. It looks like these labels were removed in the current figure. Also, Fig 1 says that panel a has a green triangle, but it's a grey triangle.

We corrected these mistakes.

Line 98: what is an on/off mesostate?

We moved an explanation of "mesostate" into the Introduction (Lines 78-81) and explain ON and OFF mesostates especially in the section entitled "Dwell-time density functions are peaked" (Lines 224-241).

Figure 1 legend says "box contains 100 binary time series" but it looks like the lower right box contains many fewer time series.

Many of the lines in the lower box pertain to cells that remained inactive during the observational time window. This gives the incorrect impression that the box contains fewer time series; it does not. We have amended the figure legend with the aim to prevent this misunderstanding.

Fig 3B: upper and lower panes are confusing – all the points are zero on the Y-axis so where did the blue lines come from?

The dots (points) on the X-axis of Fig. 3B mark the position of specific alpha-values for which mesostate dwell time densities were plotted in Fig. 3C. We have revised the figure legend to avoid any possible confusion.

In the formula on line 94, c_1 , c_2 and $\lambda_{1,2}$ are not defined and explained as far as I can tell. This is a major problem as $c_{1,2}$ are brought up multiple times in the manuscript.

Lines 136-138 (revised manuscript) state: A biexponential survival curve implies a density function of the same form: $f(t) = c_1 e^{\lambda_1 t} + c_2 e^{\lambda_2 t}$ where $\lambda_1, \lambda_2 < 0$ and c_1, c_2 are constants, three of which, λ_1, λ_2 and c_1 , may be independently chosen for fitting to the data (*cf.* Methods).

Line 100 – c_i is not defined; c_j not defined on line 105.

The indices i and j were used as placeholders for 1 and 2. In the revised text we don't use this formalism any longer.

I don't understand how the data support burst runs (line 190 and following)

Our analysis shows that bursts (ON periods) were temporally clustered; clusters were separated by OFF periods that, on average, were markedly longer than OFF periods within clusters. The revised manuscript, hopefully, makes this clearer (*cf.* section "Activator and chromatin remodelers control the length of external but not internal OFF period," Lines 220-234.)

line 211: the authors don't explain well how their data strongly supports a kinetic proofreading mechanism – the main point of their paper.

This question is now discussed explicitly in the last paragraph of the Discussion (Lines 278-292).

I cant follow the logic in the paragraph starting on line 103.

The entire section has been completely rewritten to clarify the argument (Lines 138-141 of the revised manuscript).

Line 166: what is a *pho5* chromatin ring?

For isolation, *PHO5* was released in circular form from its chromosomal locus by site-specific homologous recombination. This was explained via reference to earlier publications. To avoid an explanation in the text, we now use the phrase '*PHO5* molecules' instead of '*PHO5* rings.'

Reviewer #3 (Remarks to the Author):

1) The observed peak in dwell time distribution indicates that the underlying gene expression dynamics operates out of equilibrium with energy dissipation. The illustrative Fig. 1A seems to suggest that the source of energy is ATP hydrolysis. The natural question is how the system uses the ATP hydrolysis energy to drive the reaction cycle as shown in Fig. 3D, i.e., what is the molecular mechanism for the dissipative gene expression process?

We revised the Introduction to restate in one paragraph (Lines 62-73) what was argued in the earlier text in, perhaps, a somewhat scattered manner, namely that ATP-dependent chromatin remodeling drives the process away from equilibrium. This we now also stated upfront, in the abstract.

2) A related question to the previous one is what are the functional benefits to carry out such a dissipative gene expression process. For example, in other recent studies of nonequilibrium biological systems such as sensory adaptation (Lan et al, Nature Physics 8, 422–428, 2012) and synchronization of molecular clocks (Zhang et al, Nature Physics 16, 95-100, 2020), a strong case has been made in terms of a general tradeoff relationship between the energetic cost of the systems and their performance benefits. The question here is whether the nonequilibrium gene expression mechanism, which costs chemical energy, has any functional benefit to the system?

It remains unknown what the benefits are. However, on theoretical grounds we argued earlier, and in the Introduction of this manuscript, that the benefit must be an increase in the specificity of transcriptional regulation; the promoter preferentially responds to activators that bind specifically rather than non-specifically to its regulatory sequences. (See also revised Discussion, last paragraph, Lines 278-292.)

3) Lines 137-144: The authors say the OFF period distribution is also strongly peaked suggesting strongly irreversible entry and exit. But then say 50% of bootstrap densities reduced to exponentials and use a single microstate to model the OFF state. I found

this contradictory: it seems like the measurements do not consistently support a strongly irreversible process for the OFF state.

There is no contradiction. For instance, the process depicted in Fig. 3A with $\alpha = 0$ is irreversible, although one mesostate (State 3) entails only one and not two (or more) microstates. At the microstate level, all dwell-time densities are exponentially distributed regardless of whether the process is reversible or irreversible; non-exponential dwell times are the result of observational coarse-graining. However, fast transitioning between the microstates that pertain to the same mesostate often obfuscate the existence of multiple microstates — *e.g.*, when the density function is peaked but the peak is close to $t = 0$. All we state is that for internal OFF periods the distinction between microstates was less clear than for internal ON periods (*cf.* Lines 173-181). This is entirely unrelated to the question of whether the process is reversible or irreversible.

4) In the model (Fig. 3 B-C) the density becomes monotonic at a particular entropy production level. Does this level have any biological or theoretical significance? Can the model be used to (at least roughly) bound the entropy production level in the real system?

The answer is yes, *cf.* Skinner and Dunkel (2021), which we now cite in both the revised manuscript and the SI.

5) Fig. 1A provides useful intuition, but the finding in this paper is that the ON state has the irreversible multi-state structure. Is there a similar biological interpretation of the two ON microstates?

Like the OFF state, the ON state may have multiple molecular manifestations. For instance, the promoter may be occupied by different activators, specific and non-specific, in different binding configurations. We prefer to refrain from identifying microstates of the hidden Markov process, invoked to explain observed dwell time distributions, with defined molecular states. For the invoked Markov process with coarse-graining provides a model for our observations and not the underlying molecular dynamics (*cf.* Discussion). It is common to make this identification; and implicitly, we have done so, too: by representing both states of the underlying molecular process (Fig. 1A) and states of the hidden Markov process which models our observations (Fig. 3A and D) by the same symbol, a rectangular box. In Fig. 3, we now use a circle instead of a rectangle.

6) The difference between “internal” and “external” OFF timescales does indicate the presence of both short- and long-lived OFF states. But the criteria for separating these (whether the OFF period intersects the boundary of the observation window) can conflate the two: shorter dwell times in the long-lived state may be classified as “internal” and longer dwell times (or that start near the end of the observation) in the short-lived state may be classified as “external”. Can this effect bias or otherwise influence the measured statistics?

Yes — however, our observation that external OFF periods were purely exponentially distributed with a mean markedly different from that of internal OFF periods suggests that this bias was

negligible; since external OFF periods were also long relative to the observational time window, the process almost never returned to an ON state within the observational window after leaving the ON state for a stable OFF state (invoked to model external OFF periods). The boundaries of the observational time window, therefore, almost always fell into the lifetime of the stable and not unstable OFF state.

7) Theorem 1 in the SI assumes the eigenvalues of the restricted transition matrix $W_{\{AA\}}$ are real. But this is not generally true for non-equilibrium systems. The exponential decomposition still holds, but for complex eigenvalues with negative real part.

This is correct. However, for processes on 3-state graphs (Fig. 3A), it can be proved that the eigenvalues are always real (and nondegenerate). We now include a proof of this result in the Supplementary Information (*cf.* Part II).

Minor issues:

Fig. 2 & Extended Data Fig. 3: might be useful to mark the peak of the distribution instead of (or in addition to) the median.

We added vertical lines to indicate the position of density function maxima.

Caption describing Fig. 3D missing, it is unclear what the mutant labels in this panel refer to.

The lacking description has been added.

Fig 4 caption is wrong: should say "external OFF periods"

This is correct. The error has been fixed.

References missing in SI?

In the submission, SI references were included with the references for the main text. We now have separated the references.

SI Footnote 1 formatting typo

This has been fixed.

SI pg. 2: "diagonal elements of W are contained in the diagonal matrices $W_{\{AA\}}$ and $W_{\{BB\}}$," these matrices are not diagonal

Our statement is false indeed. The correct statement is: "Note that the diagonal elements of W are the diagonal elements of W_{AA} and W_{BB} ." (*cf.* revised SI).

REVIEWER COMMENTS

Reviewer #1 (Remarks to the Author):

I appreciate the author's efforts to make the manuscript more comprehensible. However, after reading the response and manuscripts I felt that there were no serious attempts to address my criticisms. I therefore have more or less the same reservations as I had during the first round of review. I will elaborate on this below:

- The authors argue that irreversibility of the underlying process follows from irreversibility of the observed process and I agree on this as long as the underlying process is Markovian. This is what the newly added proof of Theorem 5 shows, which I appreciate. However, from this I still don't see how this result generalizes to scenarios where the underlying process is truly non-Markovian. So if the observed dynamics can be described through coarse-graining of a Markov process (which seems to be the case), then I can argue that a corresponding underlying Markov process always has larger entropy production but it is not obvious to me what this implies if the true underlying process was non-Markovian. It is possible that such result exists, but then the authors should cite and discuss it. The two provided references also assume Markovian dynamics, which become non-Markovian upon coarse graining.
- Similarly, I still believe that the change-point detection method requires validation. I disagree that the change-point detection corresponds to a "simple" coarse-graining as argued by the authors. Every detection method based on noisy observations has both false positives and false negatives, which can affect the inferred path statistics (and hence degree of reversibility) in arbitrary ways. It is well possible that the measurement / detection noise are negligible but without validation on ground-truth, it is impossible to know.
- It is true that some of the simplest models of promoter activation have no cycles, but there are other models that do (see e.g., models that include a refractory state). Moreover, there is a difference between making simplifying assumptions in a theoretical model of transcription (e.g., time-scale separation / quasi-equilibrium), and making claims about transcription itself. While equilibrium promoter models may work very well for some systems, they may fail for others as is discussed critically in some of the references mentioned by the authors. Thus, I still don't believe that equilibrium is a widely assumed / accepted property of promoter regulation.

Reviewer #2 (Remarks to the Author):

The authors have done a good job in answering my questions and revising the manuscript appropriately. The point below should be addressed before publication:

The data to support the kinetic proofreading model is all from a single promoter that is known to be regulated by chromatin remodeling as part of the gene activation mechanism. The authors should address these limitations as well as how general their conclusions could be for other genes. For example, only a subset of yeast genes are thought to be regulated by activator-dependent chromatin remodeling.

Minor comment:

I was unable to find the supplemental figure legends in the files supplied to me and the supplemental figures were not labeled (both in the file name or in the figure), which led to some guesswork when

evaluating the figures.

Reviewer #3 (Remarks to the Author):

Our previous comments/questions have been adequately addressed in the authors' response and the revised manuscript. We suggest the revised manuscript be accepted.

Nonequilibrium Dynamics of Transcription Initiation: Evidence from Single Gene Analysis in Yeast

SECOND POINT BY POINT RESPONSE TO REVIEWERS

In the following, reviewer comments are in Arial, our response in Palatino Linotype.

Response to Reviewer #1:

"I felt that there were no serious attempts to address my criticisms. I therefore have more or less the same reservations as I had during the first round of review. I will elaborate on this below:

- The authors argue that irreversibility of the underlying process follows from irreversibility of the observed process and I agree on this as long as the underlying process is Markovian. This is what the newly added proof of Theorem 5 shows, which I appreciate. However, from this I still don't see how this result generalizes to scenarios where the underlying process is truly non-Markovian. So if the observed dynamics can be described through coarse-graining of a Markov process (which seems to be the case), then I can argue that a corresponding underlying Markov process always has larger entropy production but it is not obvious to me what this implies if the true underlying process was non-Markovian. It is possible that such result exists, but then the authors should cite and discuss it. The two provided references also assume Markovian dynamics, which become non-Markovian upon coarse graining."

We are thankful for the reviewer's thoughtful criticism. It challenged us to think more deeply and, we believe, helped us to further improve this manuscript.

We would be happy to condition our conclusion — that the underlying molecular process is irreversible — on the Markov assumption. We think, however, this is not necessary. The proof of Theorem 5 does not invoke the Markov assumption; and neither do the proofs of Seifert (2019; *cf.* Equation 16), and Skinner and Dunkel (PNAS 2021; *cf.* Section III) — see also the revised discussion following Theorem 5 in the Supplemental Information. However, all three proofs rely on the validity of stochastic thermodynamics, specifically the assumption that entropy production is a logarithmic function of the ratios of forward and corresponding reverse probability currents, the starting point of all three proofs. Although this hypothesis is less strong than the Markov assumption, it remains a bold claim and like any other theory might be mistaken. We note however that stochastic thermodynamics, so far, has empirically proved its mettle — see for instance the elegant biophysical test of Jarzynski's equality

by Bustamante and coworkers (Liphardt et al., 2002). We have revised the discussion following Theorem 5 in the SI to explicitly state the dependence on stochastic thermodynamics.

- Similarly, I still believe that the change-point detection method requires validation. I disagree that the change-point detection corresponds to a "simple" coarse-graining as argued by the authors. Every detection method based on noisy observations has both false positives and false negatives, which can affect the inferred path statistics (and hence degree of reversibility) in arbitrary ways. It is well possible that the measurement / detection noise are negligible but without validation on ground-truth, it is impossible to know.

Change point detection was used to consistently discretize fluorescence time series (proxies for *PHO5* transcriptional activity) into two types of segment (or "states"), ON and OFF. Similarly, Skinner and Dunkel (PNAS, 2021) discretized fluorescence time series (proxies for calcium fluctuations in embryonic kidney cells) into three "metastates" by plain thresholding of signal intensities. Coarse-graining generally entails both observational limitations and decision making. The coarse-grained process is always a cruder model of reality than the original process. The art is to not remove or lose to coarse-graining the properties of interest. For instance, coarse-graining may easily transform an irreversible into a reversible process (*cf.* Introduction). (The reverse however appears not possible, coarse-graining may not convert a reversible into an irreversible process.)

The problem of "noisy observations" is, we think, a separate (although, indeed, closely linked) issue. Any observation, including ours, is subject to variation that is unrelated to what one wishes to measure. For this reason, we discretized fluorescence time series ("sample paths") by change-point detection analysis rather than plain thresholding. The algorithm detects change that persists on some operationally defined time scale and thus deliberately ignores intensity fluctuations that are "too fast." Fundamentally, however, both change point detection and plain thresholding do the same: they rectify noise, and along the way make mistakes ("false positives" and "false negatives" as you say). Noise rectification together with the mistakes it commits are part of the observational coarse-graining.

The problem, we think, is not that our conclusions relied on a Markov assumption or change point detection analysis. The fundamental problem is that the process which underlies our observations is a complicated convolution of the process of interest, the mechanism of transcriptional regulation, with ancillary molecular processes (e.g., RNA polymerization and the RNA-binding dynamics of the fluorescent protein); and that it entails, in principle, the entire observational apparatus (microscope, laser, CCD camera,

analytical algorithms, even the person who performed the experiments). What complicates the matter further is that some of the ancillary and auxiliary processes are known to be irreversible (e.g., RNA polymerization, CCD camera and laser operations, which all consume energy). What then gave rise to observed non-monotonic (peaked) dwell-time densities, a hallmark of irreversibility? Our hypothesis is that peaked dwell-time densities were generated by an irreversible regulatory process for transcription. We have tested this conjecture: by targeted perturbation of the regulatory process (via deletion mutation of the Pho4 activation domain). The mutation transformed peaked into non-peaked dwell-time densities, as expected on the assumption of our hypothesis. Alternative explanations, if not impossible, appear much less plausible (e.g., the assumption that the Pho4 mutation altered the irreversibility of RNA polymerization or the interface between the observational apparatus and the molecular process). We have amended the main manuscript and the SI information to include parts this more detailed discussion.

- It is true that some of the simplest models of promoter activation have no cycles, but there are other models that do (see e.g., models that include a refractory state). Moreover, there is a difference between making simplifying assumptions in a theoretical model of transcription (e.g., time-scale separation / quasi-equilibrium), and making claims about transcription itself. While equilibrium promoter models may work very well for some systems, they may fail for others as is discussed critically in some of the references mentioned by the authors. Thus, I still don't believe that equilibrium is a widely assumed / accepted property of promoter regulation.

We don't think that "to assume" necessarily connotes "to believe in the truth of." In the abstract, we replaced the phrase "generally assumed" with "often assumed."

Response to Reviewer #2:

"The data to support the kinetic proofreading model is all from a single promoter that is known to be regulated by chromatin remodeling as part of the gene activation mechanism. The authors should address these limitations as well as how general their conclusions could be for other genes. For example, only a subset of yeast genes are thought to be regulated by activator-dependent chromatin remodeling."

We added an additional paragraph to the Discussion to reflect on this question..

"I was unable to find the supplemental figure legends in the files supplied to me and the supplemental figures were not labeled (both in the file name or in the figure), which led to some guesswork when evaluating the figures."

We apologize. Per instruction, we had included the legends to the supplemental figures in the cover letter to the editor. We now included Extended data figures (with legends) also in the File labeled Supplemental Information (Section V).

REVIEWERS' COMMENTS

Reviewer #3 (Remarks to the Author):

In my opinion, I believe the authors' second round response addresses the remaining comments from the reviewers.